# FEDERATED LEARNING WITH VARIATIONAL AUTOENCODERS

## ABSTRACT

In this work we investigate the feasibility of using federated learning to train a variational autoencoder capable of generated handwritten digits when trained on the MNIST dataset. It was found that using federated learning we were able to train a model that produced comparable results to a centralised model, both in image reconstructions and image generations.

## 1 INTRODUCTION

Variational autoencoders (VAEs) have gained popularity in recent years as models capable of learning latent representations from complicated data, and subsequently providing the ability to generate new data from this learned representation (Doersch, 2021). They have been demonstrated to be effective in domains such as image generation (Shao et al., 2020) and time series generation (Desai et al., 2021). Data generation can be useful for building larger, more balanced datasets.

Federated learning (FL) has also gained popularity as a method to train machine learning models on multiple client datasets without data sharing. FL has been shown to produce comparable performance to centralized models training for many tasks, whilst preserving the privacy of the clients data (Li et al., 2020). The use of VAEs within FL has been investigated for tasks such as collaborative filtering (Polato, 2021) and anomaly detection (Zhang et al., 2021). However, a thorough assessment of the quality of the generated data between federated and centralized models is missing from the literature. This work hopes to provide a basis for this research.

## 2 METHODOLOGY

### 2.1 DATA

For this investigation we used the MNIST (Deng, 2012) dataset, which consists of hand written 28x28 grey-scale pixel digits. Specifically, 60,000 training and 10,000 testing images were used. To create a FL scenario, we split the testing dataset independently and identically into 4 sub groups, representing 4 FL clients. The testing data was not split.

### 2.2 MODEL

To perform latent representation and data generation, we used a standard VAE architecture built in PyTorch consisting of an encoder, latent variable sampler and a decoder. The encoder consists of multiple layers of convolutions, batch normalisation and activation. After sampling the latent variables, the decoder reconstructs the data by performing the inverse procedure of the encoder. We used a latent dimension of 200 to represent the data for this investigation. The model used $\beta$-VAE loss (Burgess et al., 2018) to train.

### 2.3 TRAINING

We trained the standard VAE model for 25 Epochs using a batch size of 128 and a learning rate of 1e-3. In the federated scenario we performed 15 training rounds, where each client was trained for 6 Epochs with a learning rate of $1e-3$. We used the *FedAvg* (McMahan et al., 2017) algorithm to compute the average parameters from a round of federated training.

For full model architecture and training procedure please see `https://github.com/hugo-dugdale/federated_VAE`

## 3 RESULTS AND DISCUSSION

Figure 1 below shows a sample of 30 inputs from the test set and the reconstructions from both the standard and federated VAE models.

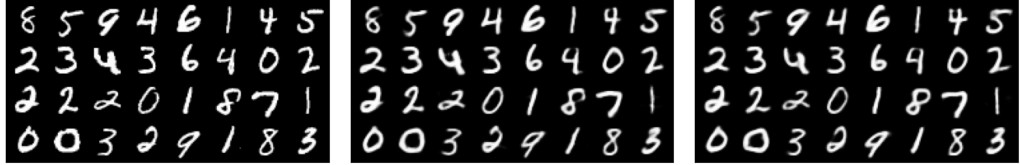

Figure 1: From left to right: input samples, standard VAE and federated VAE.

Figure 2 below shows a sample of 256 randomly generated images from the standard and federated VAE models.

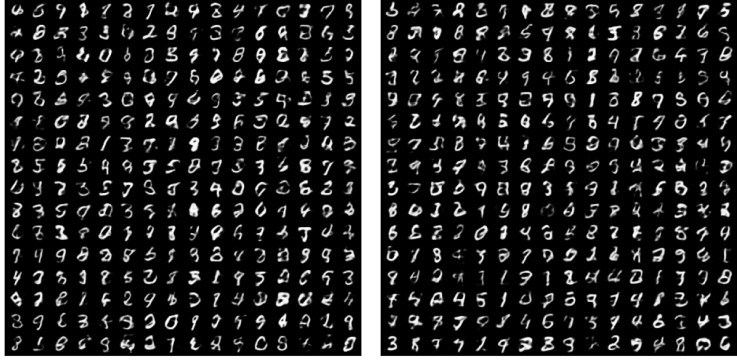

Figure 2: From left to right: standard VAE and federated VAE generations.

Figure 1 demonstrates the capability of both models to reconstruct the input data effectively. The training loss curves (see Appendix A) also show similar training patterns for both models. Figure 2 demonstrates that both models are also able to generate some realistic looking digits. However, both models do also produce digits which are unrealistic. The comparable performance of the FL model on MNIST data agrees with previous works (Nilsson et al., 2018) focused on classification tasks. Federated VAE could subsequently be used to produce a greater number of samples from underrepresented clients in a FL scenario.

## 4 CONCLUSION

In this work we demonstrated that a VAE model trained in a federated manner is able to reconstruct and generate data with comparable performance to a standard VAE model as. Further research into this area should apply more quantitative approaches to assess generated data e.g. Inception score, Frechet Inception Distance, auxiliary classifiers (Leontev et al., 2020). This investigation should also be expanded from simulations to real world federated data such as activity recognition tasks.

ACKNOWLEDGEMENTS

URM STATEMENT

The authors acknowledge that at least one key author of this work meets the URM criteria of ICLR 2023 Tiny Papers Track.

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

# Appendices

## A  LOSS CURVES

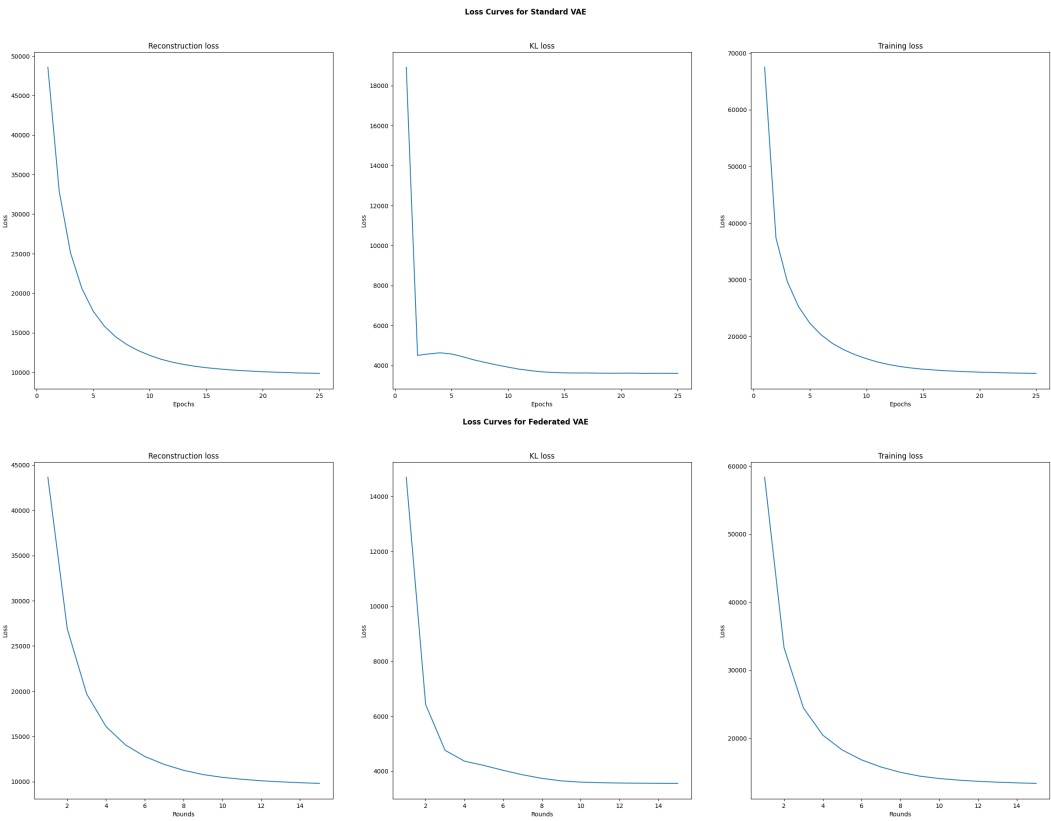

Figure 3: From top to bottom: standard VAE and federated VAE loss curves.

## B  T-SNE ANALYSIS

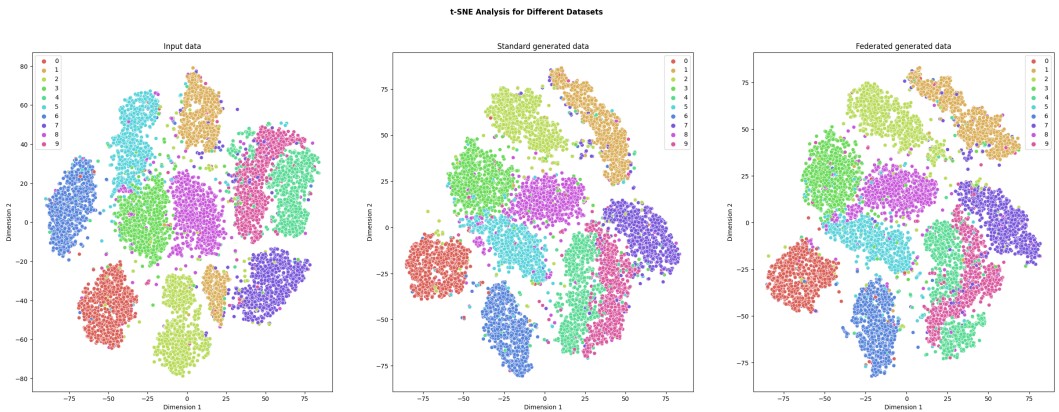

Figure 4: From left to right: t-SNE plots for test data, standard generations and federated generations.

