# OpenReview forum: "Federated Learning with Variational Autoencoders"
_ICLR.cc/2023/TinyPapers — Submitted to Tiny Papers @ ICLR 2023_

### Official Review · Reviewer_i64E · 2023-03-22

**Confidence:** 4

**Summary Of Contributions:**

Authors propose training VAE in federated learning(FL) fashion to reconstruct/generate MNIST data. Promising visuals for reconstructing MNIST data are presented.

**Rating:**

Clear, Correct, and Reproducible (CCR): a submission which meets the reviewing criteria

**Strengths And Weaknesses:**

* **Strengths**
    * An hybrid approach combining VAE and Federated Learning(FL)
    * Qualitative visuals seeming close to standard offline VAE
    * The experiments are reproducible given the code is available.
* **Weaknesses**
    * A objective comparison of quantitative metrics is missing

**Suggested Changes:**

Besides qualitative visuals having an objective comparison of quantitative metrics is invaluable to this submission. Still I would recommend this work discussions at ICLR.

---

### Official Review · Reviewer_cbM7 · 2023-04-03

**Confidence:** 3

**Summary Of Contributions:**

The paper talks about use of federated learning to generate handwritten text. It also presents code link along with generated results.

**Rating:**

Clear, Correct, and Reproducible (CCR): a submission which meets the reviewing criteria

**Strengths And Weaknesses:**

Strengths:
1. Everything is presented clearly along with the core idea
2. Results are also presented along with a discussion of future directions

Weakness:
1. Experiments have not been performed along with analysis of other methods and comparison studies.
2. Lack of quantitative studies

**Suggested Changes:**

Quantitative studies along with comparative analysis can make this a good result.
Some detailed analysis of generated images with potential ideas to improve the results can present a good study.

---

### Meta-Review · Area_Chair_3ZYx · 2023-04-06

**Recommendation:** Invite to revise
**Confidence:** 4

**Metareview:**

This work proposes a federated learning-aided VAE. This work is not solid and not interesting. Also, experiments are not convincing. The presentation is not clear.

**Summary:**

This work provides a federated learning-based VAE>

**Comments And Feedback To The Authors:**

Please review the meta review.

**Reason For Not Giving A Higher Recommendation:**

This work is not novel and not solid.

**Reason For Not Giving A Lower Recommendation:**

N/A

---

### Decision · Program_Chairs · 2023-04-08

No revision received; not invited to archive